# Current Challenges in the Adoption of Digital Visual Management at Construction Sites: Exploratory Case Studies

Ana Reinbold [1,*], Eelon Lappalainen [1], Olli Seppänen [1], Antti Peltokorpi [1] and Vishal Singh [2]

1 Department of Civil Engineering, Aalto University, FIN-02130 Espoo, Finland
2 Centre for Product Design and Manufacturing, Indian Institute of Science Campus Bangalore, Bangalore 560012, Karnataka, India
* Correspondence: ana.reinbold@aalto.fi

**Abstract:** In the construction industry, digitalisation has led to increasing efforts to improve construction management using digital visual management (VM) devices. Although the amount of research on digital VM (DVM) in the design phase and in the management of construction sites has also increased, its implementation during the production phase and by construction crews remains limited. The objective of this study is to explore the adoption of DVM in construction sites, assess construction workers' experiences regarding digital and analogue VM devices, and understand the challenges that hinder the adoption of such devices. This study used a mixed method approach, combining qualitative and quantitative research. Data included visual site explorations, surveys of construction workers and crew managers, and unstructured interviews with site managers and development directors to assess the use of DVM devices in construction sites, the need for them and their current implementation. The findings showed that VM should be conveniently located near the job site instead of the office site, which is the current situation. Construction crews who experienced more production and schedule disruptions reported that VM supported their work, compared with crews that had fewer disruptions. VM devices on construction sites are analogue, and their usage continues to be in construction site management, which perpetuates information silos during construction projects. The findings of this study provide insights into the development and deployment of DVM devices on construction sites. Construction workers' need for visual information close at hand is of interest to both scholars and practitioners in future research and development.

**Keywords:** visual management; digital VM; digital VM in construction

## 1. Introduction

Although the use of digitalisation in construction has progressed, there is a recognised lack of situational awareness (SA) in construction projects, due to information bottlenecks and the cost and time needed to collect and update data [1]. SA originated in military operations as "the perception of the elements in the environment within a volume of time and space, the comprehension of their meaning and the projection of their status in the near future" [2]. The first application of SA in construction projects focused on safety management [3], but the term can be applied to encompass the entire production phase of a construction project. Important elements of SA include improving project perception through information sharing and increasing the transparency of processes and flows on construction sites [1–4]. The lack of SA results in inefficient decision-making processes and unsustainable patterns of production that waste time, money and resources [1,5].

Construction flow refers to the process-oriented construction management. This is important because flow highlights the waste of time from the perspective of construction products and crews. Wasted time is manifested as time spent on non-value-adding activities, such as inspections, waiting or moving from place to place. Crews wait for materials, information, equipment, space, completion of preceding activities and so on, while tasks wait for crews [5].

This wasteful process presents a major opportunity for sustainability. According to the United Nations (UN), the construction industry is responsible for 40% of the global energy use, nearly 40% of waste and 30% of energy-related greenhouse gas emissions [6]. Decreasing wasted efforts by increasing SA and working towards the development of digital tools that empower construction crews during the execution of their tasks, can be linked to the UN Sustainable Development Goals (SDGs): SDG 8, creating decent work; SDG 12, using time during construction responsibly; and SDG 11, the improvement of industry practices in the creation of more sustainable cities and communities. Technological development has the potential to address this problem through real-time data collection and information sharing in construction projects. Exploring the digital collection of data in construction sites from various sources and systems [7–12] and making them available as meaningful information for construction workers, can also improve their awareness of the construction site and the current production status, as well as increase their autonomy during the decision-making related to the task development [13]. However, although much research has been conducted on the digitalisation of the construction industry during the last five years [14], most studies have not focused on workers. Digital visual management (DVM) could fill this gap by making SA information available during production activities on construction sites. VM aims to provide visual information that can be immediately retrieved by workers and immediately transferred to task execution [15]. By increasing the visibility of the production flow, VM increases transparency, as a process can be communicated by and to all participants [16,17]. To harness this potential, an understanding of the different information needs of the actors present on construction sites is necessary.

However, the worker's view has been neglected, and the adoption of VM in construction sites has supported mainly managerial decisions [18,19]. Most VM implementations in construction are individual tools that support specific operations, such as budget follow up, general schedule visualisations and safety requirements, without considering the need to support production activities in all phases [6,20–22]. In addition, the process of devising a VM device is complex and time-consuming; moreover, it requires an understanding of the users' information needs and a plan for updating information [23]. Research has focused on the implementation of DVM during the design phase and on upper management on construction sites [24–28]. However, there has been little research on the implementation of digital VM devices in the production phase of construction projects or on sharing information with construction crews.

This study explored the status of the implementation of DVM on construction sites. Regarding its methodology, this study utilised site observations to address the types and locations of VM device implementation, surveys of workers and crew supervisors to understand their previous experience with VM and the correct positioning of VM devices on construction sites. Unstructured interviews with site managers and development directors were conducted to assess their understanding and support during the adoption of VM. The aim was to evaluate the implementation of DVM devices.

The results of this study showed that in addition to efforts to increase digitalisation in construction [29,30], the digitalisation of VM devices on construction sites followed the model and logics of analogue VM devices, which were used in trailers and office areas. Moreover, the information was distributed only to site managers. The results also revealed that when construction crews faced time schedule challenges, they wasted time searching for information that was located away from the work location.

## 2. Materials and Methods

The study design consisted of multiple case studies and construction companies located in Finland. The aim was to explore the context and the activities and to identify problems related to the adoption of DVM in construction [30–33]. The case study method was chosen to enable the researchers to explore the current status of the adoption of DVM in current construction projects, by collecting data and observing phenomena in situ. In the case studies, data were collected by observations, interviews and surveys.

The three investigated cases are described in Table 1. These sites were selected, based on the following criteria: (1) company adoption of VM devices; (2) willingness of the construction companies to participate in the study; (3) free access to researchers' site observations; (4) opportunities for interviews with workers and site managers; and (5) opportunities to survey construction workers, regarding the implementation of VM devices [34].

**Table 1.** Case Descriptions.

| Case | Area (m²) | Project Type | Investment (EUR M) | Project Duration (Years) | Construction Company Size (Yearly Revenue EUR M) |
|------|-----------|--------------|--------------------|--------------------------|--------------------------------------------------|
| **A** | 75,000 | Infrastructure | 250 | 3 | 932.55 |
| **B** | 4000 | Multi-purpose | 100 | 3 | 932.55 |
| **C** | 135,500 | Commercial | 170 | 4 | 17,284 |

Cases A and B were by the same construction company. This company implemented VM devices in construction management and provided information and access to two different projects.

In case C, visual devices were implemented to support takt production. Takt production is a production planning and control method that focuses on the identification of repetitive processes in production and establishing production flow to increase their efficiency [35,36]. Case C provided the opportunity to collect workers' opinions about visual management. The implementation of VM devices, focused on one area of the site. Another area where no VM device was implemented was used for comparison. The implementation occurred during the COVID 19 pandemic, and site access was restricted to workers. Therefore, the leading researcher was not able to visit the site. Site personnel documented the use of VM devices through photographs, which were made available to the researchers.

Site observation visits (cases A and B) gave the lead researcher the opportunity to observe real construction sites and all visual management devices used in the project. The benefit of the observation approach is that the researcher's participatory observations and time spent with employees and management, facilitate a deep understanding of the phenomenon [37]. This method also allows the researcher to approach the group under study, as closely as possible. The observations were used as a "probe" before the interviews with the construction workers and management to improve the relevance of the interview questions. Further information about the observations was evaluated using the data collected in the interviews [38].

In this exploratory study, the interview method was used to gather data using two different approaches. Structured surveys were conducted with construction workers, whereas unstructured interviews were conducted with site managers and development directors [39]. The survey method was applied in this study to assess the previous experiences and opinions of the VM of both construction management and construction workers. Surveys were conducted with workers to limit the time required to disrupt their activities and to support the comparison of the results from the groups of construction workers [40]. A researcher was available during the surveys to explain the survey and its contents to the crews, as well as to clarify the workers' doubts. In contrast, the interviews with site managers and development directors were unstructured. The themes were based on VM knowledge and understanding, with the aim of generating qualitative data by asking open questions and allowing the respondents to freely communicate their thoughts and opinions [41].

Regarding the qualitative validity and reliability, the research design included methods, by which the accuracy of the findings was assessed. Moreover, the steps in the research process were conducted in a consistent manner [42,43]. Regarding reliability, the methods used in this study included a cross-examination of the data by the authors. In addition, a common database was created, in which the raw data collected in the study were stored

and processed. The data were also triangulated [44]. The data collection and analysis are described in the following sections.

### 2.1. Data Collection

The site exploration was based on participation and observations, which resulted in field notes and photographs prepared by the lead researcher [32]. At this stage of the research, accompanied by the site managers, the lead researcher walked around the site and photographed the VDM devices [44]. To ensure the data protection and privacy of the site personnel, the photographs were of only visual devices, and the presence of individuals was avoided. In addition to photography, the researcher also systematically observed visual phenomena, related to VM as well as phenomena that were missing from the site. During the interviews with site managers and the directors' sessions on development and implementation, the researcher asked many open-ended questions that were answered by the site staff. The field researcher recorded these findings in her field notes. The survey of the workers was conducted at site C.

The site exploration focused on the implementation of visual devices, their location on the construction site, the target group of the device, the type of information shared and whether the means of sharing was digital or analogue. An observation protocol was designed to guide multiple observations in this qualitative study, following research methods, discussions and McLeod's recommendations [42]. The observations focused on the following: (1) use of visual devices; (2) location and target group of visual devices; and (3) type of information shared and the method of sharing.

The photographs collected from the research sites were particularly well suited to the study of VM phenomena. The photographs enabled the visualisation of meanings, characteristics and context, which enhanced the researchers' interpretations of the phenomenon [44]. In this study, the photographs served as visual evidence that contributed to confirming or rejecting the assumptions and claims of the study [45–47]. The photographs were also used to validate the interview responses and the open-ended responses recorded in notes during site visits [45]. In cases A and B, photographic sessions were planned during the site visits after the interviews were conducted with the site managers and development directors. In case C, photographs were taken by site personnel and then analysed by the researchers. Photographs were taken of the identified visual devices, including any VM tools improvised by the workers.

The survey of the workers was conducted at site C, where VM devices for takt were implemented. The survey was conducted both in the implementation area, where a VM device was implemented, and in the control area, where no VM device was implemented. The survey process used in the present study was adopted from Grönvall et al. [34]. A short questionnaire with five questions about the use of VM devices was distributed. Three questions were binary (yes or no), one question was multiple choice, and another question was binary (i.e., production area or trailer area). To ensure consistency in the data collection, a survey protocol was developed, which included the questions to be answered by the workers, the identification of the worker's crew and in which week the survey was conducted. The surveys were conducted during calendar weeks eight and 13 in 2021, and a total of 35 workers participated. The answers were anonymous, and participation was voluntary. However, the aim was to select workers from different contractor sites and, where possible, from different areas. The survey protocol questions are shown in Table 2.

**Table 2.** Worker and Crew Supervisor Survey Protocol.

| Question | Answers |
| --- | --- |
| Have you used VM devices before? | Yes/No |
| Do VM devices support your work? | No/Very little/Little/Supports/Strongly support |
| Where should VM devices be located? | Production area/trailer area |
| Would you like to have more VM devices? | Yes/No |
| Do you believe that you can help to create VM devices? | Yes/No |

To evaluate the success of VM devices used at site C, the monitoring and controlling of Takt schedules was also conducted. Activities were tracked to identify delays, both at their beginning and at their conclusion. Then the percentages of delayed activities and punctual activities were calculated and a follow up of the percentages took place.

The interviews with site managers and development and implementation directors were conducted on sites A and B. These interviews were unstructured and based on open-ended questions. The interviews focused on the interviewees' knowledge about VM, their engagement in adopting such devices, challenges in implementing these devices and their expectations regarding the results of implementing the VM. The planned duration of the interviews was 90 min, which allowed enough time for the interviewees to discuss and formulate their answers. The lead researcher took handwritten notes on the answers and discussions, which were then compiled into a digital file after the interviews.

### 2.2. Overview of the Data Analysis

Following Creswell [45], the data were aggregated into three groups: the sites; the interviews with site managers and development directors; and the surveys of the construction workers and crew leaders. In the first stage of the data analysis, the photographs were reviewed. To ensure anonymity and the protection of the participants' personal data, photographs that included images of individuals were discarded. The interview notes were reviewed to identify incomplete and misleading information and inconclusive notes, which were removed from the final data set before the analysis. During this review, in notes from the interviews with the site managers and development directors, it was possible to identify important remarks that were then highlighted. The responses to the surveys conducted with the construction workers were scanned, and these data were stored.

The interviews with site managers and development directors and site observation protocols were analysed in combination. The aim was to identify similarities and contradictions between the points discussed during the interviews with the site managers and development directors and the use and adoption of VM devices in the different site areas, trailers and office areas and the production and construction areas.

The photographs were processed and analysed to identify the implementation of VM devices, the target group, the type of information shared, their location and status and whether VM was planned or improvised. The images were analysed with the site observation protocol and were found to support the notes taken during the visits regarding the use, location, the group targeted to receive the information and the type of information shared.

The data collected from the surveys with the construction workers and crew leaders were statistically analysed by focusing on the crew, the answers given and the median and standard deviations. The surveys were conducted on site during working days. In total, 35 employees at site C answered the survey: 20 in the study area where VM devices had been installed and 15 in the comparison area.

Sites A and B belong to the same construction company and the data collection on both sites was conducted, according to the same methodology; thus, the data sources were linked because they reflected the internal understanding and sharing of knowledge by the site managers and development directors, which emerged during the interviews. In addition, the photographs taken at both sites showed similarities in the approach to the

adoption and usage of VM among the construction workers. The data collected from site C included a survey regarding the adoption of VM devices. The responses added the construction crews' perspectives regarding VM and its implementation to the pool of data collected in this study. The data sources and their connections are presented in Table 3.

**Table 3.** Presents all data sources utilised during this research and their inter-connections.

| Case Study | Linkage to Other Data Sources | Analysed Aspects | Data Source |
|---|---|---|---|
| A | B | Knowledge from managers about VM Location and target group of visual tools | Two interview notes (site manager, 15 years' experience in construction sites, and development director, 20 years' experience) One site observation visit Fifteen photographs |
| B | A | Knowledge from managers about VM Location and target group of visual tools | One interview note (development director, 20 years' experience) One site observation visit Fifty-five photographs |
| C | | Survey with construction workers and crew leaders, photograph analysis, VM | Thirty-five survey responses Two interview notes (site engineer, 5 years' experience, and site intern, 2 years' experience) Thirteen photographs |

## 3. Results

This section presents the findings from the construction site visits, the analysis of the data collected in surveys with the workers, and in unstructured interviews with the site managers, development director, site engineer and site intern. The most interesting finding was that although the site managers and development directors recognised the importance of visual management for increasing transparency and improving construction flow during task execution, the devices, both analogue and digital, were still located in the trailer area for the benefit of upper management at the construction site.

### 3.1. VM Implementation Status

At construction site A, an overload of visual information was displayed inside the construction management offices, as shown in Figure 1. There was no clear identification of the date of creation or the version of the VM device. Moreover, it was not possible to identify the group targeted for using the VM devices. While information about the VM devices was plentiful in the management office areas, which were close to the production areas, they focused on health and safety matters. VM devices focusing on production were improvised by construction crews, such as marking them with different colours on slabs or placing written messages near electrical installations. The only DVM device was located inside the office area.

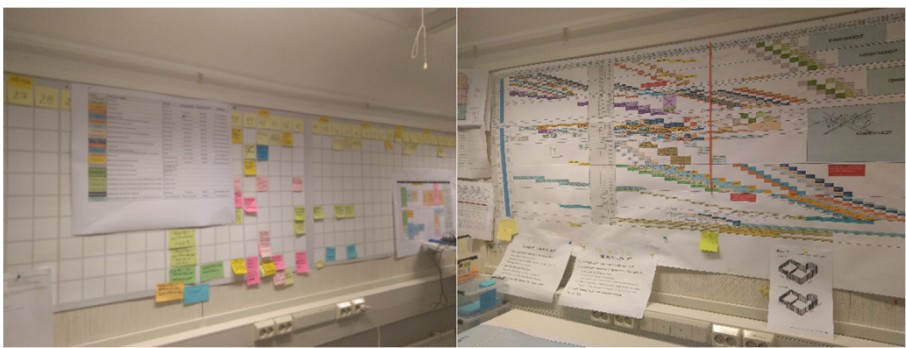

**Figure 1.** Visual management devices in the trailer and office areas in site A.

In construction site B, information was displayed in the construction management offices but without a clear target, version or date of the VM devices. The office area was crowded with VM devices, and the DVM adopted was located inside the construction management office. Images of the VM devices on site B are shown in Figure 2.

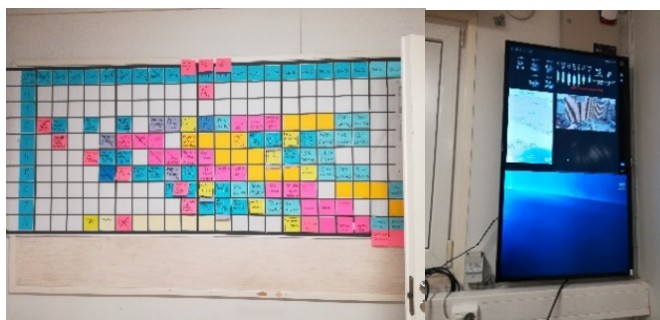

**Figure 2.** Visual management devices in the trailer and office areas in site B.

In construction site C, the implementation of VM devices was related to takt scheduling, including the takt schedule on a takt board near the production areas, takt cards with different colours for each crew and the marking of takt areas on site. The cards included information, regarding site preconditions, tasks inside the takt, the resources needed for the wagon, inspections in the wagon, and the handover date. Following the iteration with workers feedback, a three-week lookahead plan that contained information about the next two takt wagons in the takt area was added. The three-week lookahead plans were placed beside the takt cards, near the entrance to the takt areas. The following iteration considered increasing the visibility of the takt cards. This iteration increased the visibility of the takt cards because they were larger and illuminated by a lamp [34]. The VM devices implemented on site C are shown in Figure 3.

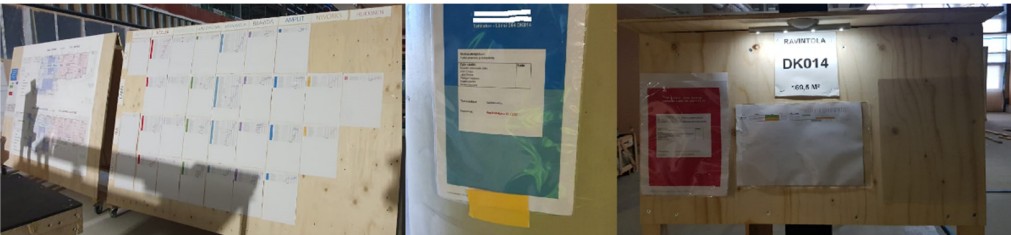

**Figure 3.** Takt cards implemented on site C fixed near the task execution area.

Several visual devices were identified at all sites, which demonstrated a willingness to share information and increase SA. In particular, in the construction management offices, a great deal of visual information was displayed on the walls, which was visible to everyone inside the office. However, most of the information did not include an identifiable version or date of use. In addition, several versions of the same information on visual devices were visible, but the currently valid version was not clearly identified. Furthermore, the information conveyed on the visual devices did not always have a clear purpose or a defined target group. Similarly, the VM devices lacked definitions, reflecting that their development was still in the early stages.

In the trailer and office areas of all the sites visited, there were visual devices showing schedule, budget, quality, progress, weather and safety. The devices were fixed, and their functions were limited to displaying actual budget and schedule progress. The target group for these visual devices was the site management team and the contractors' management team. At sites A and B, digital visual management devices were implemented. These were also located inside the trailer and office areas.

In the construction areas in sites A and B, the formal visual devices were implemented only to disseminate information about safety, such as reminders about the use of personal protection equipment, barriers to avoid falls, delimitation of risky areas, and suspended load areas. No information was displayed near the location where the construction activities took place. If the crew working on partition walls needed to re-plan their work, they did not have information about priorities, other crews' work or the impacts of their activities on other crews. This led to the lack of SA in decision-making and increased improvisation. Information had to be sought in trailer areas distant from the production area. In site A, this distance was an average of 200 horizontal metres; in site B, the average horizontal distance was 80 metres and an average vertical distance of 50 metres. Site C was notably different from sites A and B because the development and implementation of a VM device targeting production crews had been specifically tested in that project.

At the three sites, it was possible to identify improvised visual devices that had been created by the construction crews to communicate with each other (Figure 4). Crews had painted tags on the floors in distinct colours to mark the locations of different drywall structures, drawings for the placement of electrical boxes and information about levels and alignments. These informal, or improvised, VM devices appeared to have been used to communicate and present information on the construction site as part of the task performance.

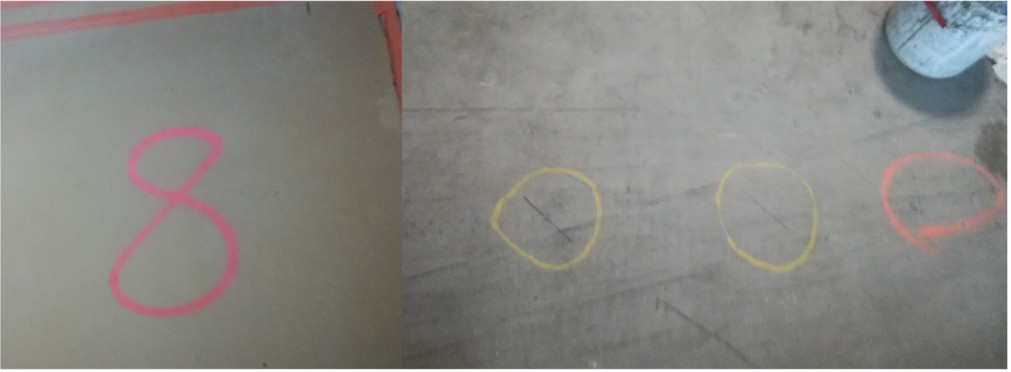

**Figure 4.** Improvised VM communication painted on the floor, close to the task execution area.

DVM devices were implemented at sites A and B. At both sites, the devices were situated inside the trailer and office areas and did not include information about tasks or production. In site B, the DVM device focused on providing real-time weather forecasts. The DVM device in site A displayed comparisons between planned and executed project budgets, as well as health and safety key performance indicators (KPIs) regarding the number of accidents and incidents on the construction site. This information was collected manually and updated monthly, which resulted in the sharing of outdated information. The site manager in site A stated that the information was scattered among different systems and that producing a monthly report required more than three weeks of data collection and analysis. Table 4 presents the main findings, according to the case study.

### 3.2. Information Needs

Sites A and B belong to the same company, and the site managers and development directors interviewed have an enthusiastic attitude related to visual management and the adoption of visual devices. The company is also interested in applying the Lean construction philosophy. Among the answers about the company's interest in visual management and its adoption, the development director said that "visualisation is the key to collaboration" and "we want to use VM to increase transparency and trust", making it clear that the general understanding of VM and its features is present in the company and in the project.

**Table 4.** Main Findings of the Case Studies.

| Findings | Case Study |
|---|---|
| Overload of VM devices in the trailer area | A, B |
| Display of outdated information | A, B & C |
| Lack of VM devices related to production near production areas | A, B |
| VM devices improvised by work crews | A, B & C |
| Digital VM updated manually and located inside trailers | A |
| Digital VM digitally updated located inside trailers | B |
| Implemented VM devices related to production located near the production areas | C |
| Manually implemented VM devices | A, B & C |
| Only 14% of crew workers had previously utilised VM in their work | C |
| Workers' need for updated information | C |

In site A, the site manager recognised that there was a synergistic connection between VM and Lean methodologies, stating, "We want to use VM to make people who never heard about Lean, work in a Lean way". He also highlighted that the increasing complexity of projects called for a more collaborative way of working. Here, there was input and support from the development director, who observed a meaningful difference between cooperating, what he described as accepting the culture and manners of work and collaborating, when the parties were involved in developing the culture and working together.

At site C, the surveys assessed the construction workers' opinions regarding the implementation and use of VM devices. Surveys were carried out during site visits, which were limited by COVID-19. All workers working in the areas of the study were invited to participate. Around 85% of workers agreed to participate. The exact number is unknown, due to COVID-19 and the use of masks and the dynamic nature of the work environment, it is possible that the same person was asked twice or some person was ignored. The trades of the survey respondents are presented in Table 5.

**Table 5.** Number of the Survey Respondents Per Contractor.

| Work Phase/Contractor | Number of Survey Respondents |
|---|---|
| Masonry | 6 |
| Water and sewage pipes | 6 |
| Electrical installation | 5 |
| Partition walls | 4 |
| Ventilation | 4 |
| Fire protection | 1 |
| Painting | 6 |
| Automation | 2 |
| Sprinkler installation | 1 |

The crews' experiences in using the VM devices are presented in Figure 5. The mean, median and standard deviations were calculated for the results of the surveys, and the averages were calculated for both the study area and the comparison area, as well as the total number of interviews. The most significant finding was that there were major differences in the groups of workers who wanted more visual tools. In contrast, the results show resistance by painters and partition wall workers to any VM. This result also reflects the low levels of willingness to participate in the elaboration of visual devices. Only five workers (14%) reported having used visual tools in the past. It is also noteworthy that the

workers were not given training or detailed information about evaluating the methods on site.

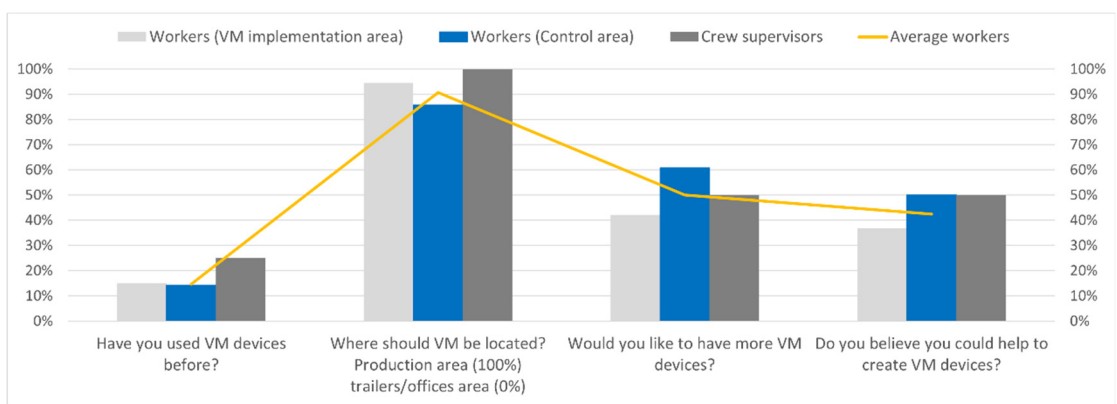

**Figure 5.** Survey answers by the crew.

The construction workers stated that visual devices supported their work on site. One crew supervisor had used visual tools in the past and was optimistic about their potential. It is also worth noting that the heating, ventilation and air conditioning (HVAC) crew supervisor commented that such methods were usually available only to site management, and it was a "big plus" when they were also used on site.

The results of the workers' survey showed that only 14% had used VM devices in previous work, and 50% stated that they would like more VM devices related to their tasks. A similar number of workers believed that they could help create VM devices that were focused on task development.

Figure 6 shows the results of the surveys by workers, the area of study and the crew supervisor.

**Figure 6.** Responses of the workers and crew supervisors to the survey.

As shown in Figure 7, VM device tools supported work was more negatively perceived than in the comparison area, from 2.76/5 to 3.21/5 (median 3.0 vs. 4.0). This may have been due to the initial version of the production plan cards that were implemented, which were considered small and difficult to find.

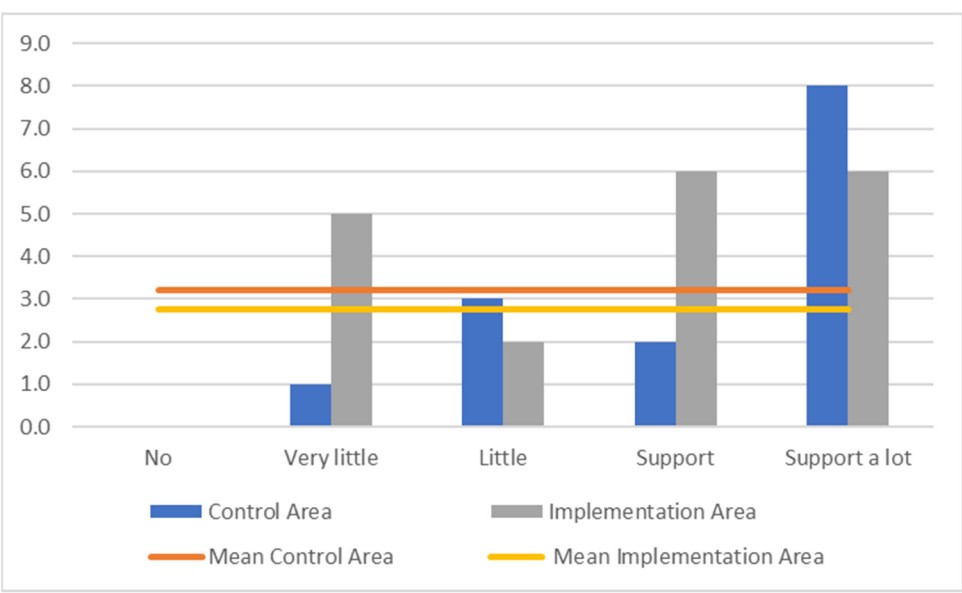

**Figure 7.** Survey responses to the question "Does VM support your work?".

In addition, the groups of workers, such as the partition wall workers and the HVAC installers, who felt that they received the least help from visual work management, also experienced the least disruptions and delays on site. In contrast, the electricians, who had suffered the most from start-up delays and disruptions, felt that they benefited from visual aids. Exceptions to this trend, in which perceived disturbances had increased the perception that visual tools were useful, were the plumbing contractor's workers, who considered the visual tools useful, even though their work had gone well.

In this study, 91% of the construction workers and 100% of the crew supervisors considered the construction site the most effective location for disseminating information as close to the workplace as possible. This finding echoes the assumption of this study, as the information was specifically intended to be implemented on the site. Nevertheless, at construction sites A and B, the visual devices were kept and displayed only in the trailer and office areas. In site C, VM devices were located near the production area. In this implementation case study, the devices mentioned were exclusively those that were part of the implementation test.

### 3.3. Current Challenges

Although the site management recognised the need for shared SA during task execution close to production activities, there was a clear contradiction between the statement by the site manager of site A and the results of the researcher's observations at the same site. Although the site manager wanted to use VM to make people who had never heard about Lean work in a Lean way, there was no use of VM devices near the production areas, and the formal VM devices in the production area were related only to safety and safety equipment.

At sites A and B, the site managers and development directors mentioned several challenges in implementing new managerial approaches, such as Lean or VM. Successful implementation depended on the involvement of stakeholders and support from upper management.

The findings showed that cultural challenges were abundant. A common approach was to make information available only when specific information was needed. Information was stored or displayed away from the production areas, and workers received information only when they required it or were unable to find it. This finding was related to the concentration of information in site management areas. In two of the three cases, production-related information was not displayed near the production areas, and production information needs were neglected.

The culture of pushing information was still strong in the construction sites, so the display of information without an immediate need for it was also observed in the three sites. It was possible to recognise a consistent practice of displaying outdated, duplicated and purposeless information. This finding indicated a misunderstanding of the difference between the availability of visual information and visual pollution, or excessive information that can impair the ability to identify valid information and directly apply it during task execution.

Regarding the digitalisation of VM devices, only one device each was implemented in construction sites A and B, both of which were located in the construction management offices. Only at site B did the DVM device display automatically updated information, which was focused on weather forecasts. In site A, information collection and DVM updating were carried out manually. The lack of automated data collection and VM systems resulted in the sharing of outdated information and the lack of SA in both the production areas and the site management offices. The potential of DVM devices to increase SA among the production and management of the construction remained unexplored.

*3.4. Results Synthesis*

Combining the responses from worker interviews, site managers and development directors with the researcher's on-site observations, led to the finding that the formal visual tools in the case studies were applied by management to management regarding the project schedule, budget and progress overviews and from management to workers regarding safety in the site. The workers were not involved in planning and implementing these devices. Nevertheless, the workers improvised VM devices during the production phase, and they recognised their importance in creating visual communication. The DVM devices were not focused on the production phase, and they were located away from where the information was needed. The DVM devices followed analog logic, where the data were collected and treated manually. The DVM device was used only to display information.

## 4. Discussion

The benefits of visual management have been extensively discussed in previous studies in the literature on production management [18]. The present study demonstrated that, in the construction industry in Finland, the elements and devices, both digital and analogue, used in visual management were kept in the trailer and office areas and targeted site management; moreover, the increased adoption of DVM had not yet impacted the construction sites [18]. Given the original goals of visual management to improve employee engagement with the company's current strategic and tactical goals, visual management in construction sites is based on strengthening the control side of the traditional "command and control" chain [48,49].

In the case study where visual devices were implemented, the workers' responses regarding the usefulness of these devices differed between the area where the device was implemented and the comparison area where no device was implemented. The workers in the comparison area found it more useful to have the information than those in the VM implementation area. This finding indicated that having information and participating in the VM device implementation process, induced criticism because of the workers' better understanding of the process and the use of visual devices.

Different crews reported diverse needs for VM, which could have been the result of their different experiences in using VM devices and difficulties faced during task execution. An important finding that emerged from the interviewees' statements and the comparison of their schedules is that crews that had fewer disruptions, considered that the VM did not support their work, and they reported less need for it. In contrast, the crews that faced more schedule and production disruptions reported a greater need for VM devices and considered them more supportive of their work. It is logical that the workers who needed support tended to search for more information and thus considered that the VM was beneficial.

The distance between the trailer and office areas, where production schedules and other crew information were available, caused a considerable amount of wasted time and a lack of SA, among the construction crews. When workers needed information that was not available or near the production area, they had to stop production and go to where the information was available, which increased waste and lowered productivity [50,51].

It is important to understand that the mere display of information through a digital device does not correspond to digitalisation. The two DVM devices implemented in this study were also located in the trailer and office areas. Only one DVM presented real-time data and automated updates. The other DVM was a digital device, based on an analogue model of creating and updating information. However, the data displayed were collected manually and analysed to generate the graphs shown.

When visual information is not shared, site operations are based only on questions [52]. Workplaces generate huge amounts of data, as the findings of this study showed; however, these data need to be interpreted. According to Galsworth [52], "It is the meaning, that we are after". The findings of this study revealed that sharing information during the production phase rarely occurs in construction projects, and construction workers operate by asking questions, and they must stop production and move to another location to obtain information. Attempts to create DVM follow analogue logic, in which data are collected manually and the production of information is a slow process. Furthermore, the iterations required to improve VM devices demand time-consuming manual work to address problems that could be solved or minimised by the implementation of DVM devices.

This study contributes to the knowledge to the visual management during the construction phase by finding that the workers were in favour of focusing VM and DVM on the execution of tasks and locating them near the task area. The study also identified that crews that had more schedule disturbances considered that VM devices near their work area would provide more support for their work, compared with crews that had fewer schedule disturbances. Another important contribution of this study is related to the implementation of DVM on construction sites and how they apply the same logics as analogue VM, where data are collected manually and are available at a distance from the task areas. The practical implications of this study include the development of DVM devices to solve long-lasting construction waste and communication problems and the development of digital tools using digital logics to remedy time-consuming and costly manual data collection practices in construction sites.

## 5. Conclusions

This study's most significant finding was that in addition to the understanding of VM concepts by upper management in construction sites, the implementation of VM devices that target workers during the execution of tasks is limited, and workers prefer to fill the gap in improvising visual communication. The interview findings revealed a lack of involvement of construction workers and crews during the implementation of VM devices and that both analogue and digital information was distributed to construction site management.

The digitalisation of VM devices in construction sites followed the analogue model, placing them in the trailer and office areas and focusing information distribution on site managers.

Based on the findings of this study, three recommendations are suggested for construction sites while implementing DVM devices. The first recommendation is to avoid the current model of DVM devices that are implemented on construction sites, where they are concentrated in construction management offices. The second recommendation is that DVM devices are focused on task development on site, which would solve two problems: the availability of information for workers and the amount of time wasted in searching for information. Third, the creation and implementation of DVM devices should not be based on analogue logic, where information is collected and treated manually and only displayed in a digital manner; instead, the entire process should be digitalised. This will

provide updated information to the crews during tasks execution, increasing their SA and enabling them to make work decisions independently. More autonomous work crews allow supervisors and managers to spend more time on preparing upcoming tasks, rather than directing crews. This should improve project outcomes.

*Study Limitations*

Because the case studies were conducted in Finland, and involved Finnish construction companies, the findings may not be applied to worksites in other countries. Therefore, generalisations, based on this study's results should be made with caution. The investigation of challenges to implement DVM focused on the production phase of construction projects in other countries is encouraged and the comparison of results from different countries can generate results that can be generalized. The survey respondents were not a random sample, but were recruited through convenience sampling through employees who were present at the construction sites. Sites A and B were different projects by the same company, which was enthusiastic about the adoption of VM. Thus, these sites may not be representative of "typical" construction sites. During the surveys with workers in case C, some of them requested clarifications, and during the explanation, the researchers might have influenced the workers' answers.

Interviews with the workers were conducted during the implementation of analogue VM devices that targeted the production area. Further research on the implementation of DVM and workers' participation in the process could contribute insights into the implementation of such devices and the information needs of workers. The development, testing and evaluation of DVM devices requires both theoretical and practical research.

VM in production processes on construction sites has been adopted to differing degrees among countries, as can be recognised [17,53]. Such differences may indicate distinct levels of maturity of the companies that have implemented VM. The development and testing of a maturity model for VM adoption could support and explain such differences. Further research is needed in this area.

**Author Contributions:** Conceptualization, A.R.; Formal analysis, A.R.; Funding acquisition, O.S. and A.P.; Methodology, A.R. and E.L.; Project administration, O.S.; Resources, O.S. and A.P.; Supervision, O.S., A.P. and V.S.; Writing—original draft, A.R.; Writing—review & editing, A.R., E.L., O.S., A.P. and V.S. All authors have read and agreed to the published version of the manuscript.

**Funding:** This research was partially supported by the Building 2030 Consortium of Aalto University and 21 companies.

**Institutional Review Board Statement:** Ethical review and approval were waived for this study because no sensitive data, regarding the subjects was collected.

**Informed Consent Statement:** Informed consent was obtained from all subjects involved in the study.

**Data Availability Statement:** The data presented in this study are available upon request from the corresponding author. The data are not publicly available to assure subject anonymity.

**Conflicts of Interest:** The authors declare no conflict of interest.

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
