# Peer review of "Current Challenges in the Adoption of Digital Visual Management at Construction Sites: Exploratory Case Studies"

_sustainability, doi:10.3390/su142114395_

Round 1

Reviewer 1 Report

The article presents a study that explores the adoption of DVM in construction sites, assesses construction workers’ experiences regarding digital and analogue VM devices, and understands the challenges that hinder the adoption of such devices. The study is based on visual site explorations, surveys of construction workers and crew managers, and unstructured interviews with site managers and development directors to assess the use of DVM devices in construction sites, their need, and their current implementation.

I find the topic announced in the abstract and introduction interesting and worth exploring, and I have some general suggestions for authors.

In the title is the word "digital", and in the abstract and introduction, "Digitalisation" is presented. However, this theme in the research development is marginal, partly because it was not found in the case studies. Therefore, I think more reflection and discussion of this aspect is useful, possibly compared with other literature case studies. The limited number of case studies may also not report a representative picture of the reality on this issue, and therefore comparison with other studies I believe is essential.

The case studies chosen are all in Finland but this is not explicitly stated except in the "Discussion" and "Conclusions" sections. I think it is crucial that this aspect is stated in the presentation of the case studies and also motivates the choice to limit to a single territory.

Some concepts are repeated in the text, so I suggest a thorough re-reading to make the text more fluent and clear.

I have some specific comments:

Line 150: consider placing all units of measurement in parentheses and on the second line to make headings clearer and more uniform.

Lines 150-151, 268-269: Leave a blank line between the table and text

Line 154: consider explaining what it is the “takt production”

Line 268: Table 2 is not mentioned in the text

Line 332: Start with a capital letter

Line 453: Numerous elements are repeated in the findings of each case study; consider constructing the table differently to bring out more clearly the commonalities and peculiarities

Lines 768-781: review this part

Author Response

We appreciate your constructive comments and support to improve our paper.

Please find in the PDF attached a detail answer, point by point, to your comments and suggestions. 

Reviewer 2 Report

(1) You authors aim to explore the status of the implementation of DVM on construction sites, which is of great value.  But why did you chose the three cases? Among thses cases, two of them belong to the same company, and the three projects are  different in characteristics and field, so I want to ask if these features will influence the research results. (2) Several figures are not clear in this paper, and some of them may be not necesssary. Please check it. (3)The Discussion and results are not clear, and can be improved. (4) I suggest the authors search and add more references publised in the recent 3 years.

Author Response

We greatly appreciate your constructive and positive comments, and we thank you for helping us improve the paper. Please find in the attached PDF a detailed answer, point by point, to your comments.

Reviewer 3 Report

The sample representation needs to be studied and the novelty statement needs to be strengthened. Articles need to improve their descriptive statistical analysis and sharpen the elaboration process with SA, VM, DVM, SDGs and sustainability

Author Response

(The authors gave the same response as above.)

Reviewer 4 Report

This study investigates the challenges in implementing Visual management tools (VM) in construction projects. The topic is interesting, and the discussion section provides some insights that can support project managers in the successful adoption of VM devices. However, there are some concerns regarding (1) research methods - Materials and Methods- and (2) the contribution of this study. Please review the following comments: 

1.      Abstract:

The abstract requires to cover the presentation of the case study- Finnish construction companies- and the research method used in this research (Mixed method or Qualitative!).

2.       Materials and Methods

Location and company size are required to be clearly presented in Table 1.

(Case descriptions)

For further information regarding the presentation of the case study, you could refer to the following article: Visual Management in Brazilian Construction Companies

http://dx.doi.org/10.1061/(ASCE)ME.1943-5479.0000354

The information regarding the experience level of the respondent needs to be presented to provide a comprehensive overview of the respondent's profile in Table 3.

Further explanation regarding the quantitative analysis is needed; for example, the response rate requires to be clearly identified.

3.      I recommend the authors add a section/ subsection discussing the study's limitations with more details and the issues that should be addressed in future research based on the current study's findings. (Discussion section- Line 681-710)

4.      In the conclusion section, the authors should provide key takeaways from the results of this study rather than a summary of the paper

5.       Additionally, the contribution of this paper to the construction body of knowledge and industry practices needs more clarification.

6.      The authors should explain how the results of this study can be globally applied (for example, to the U.S. construction industry).

 Finally, I recommend this article be published in "Sustainability" after minor revision.

Author Response

We greatly appreciate your constructive and positive comments, and we thank you for helping us improve the paper quality. Please find tin the PDF attached details answers, point by point, to your comments and suggestions.

Round 2

Reviewer 2 Report

Thank you for your efforts. I have two suggestions for you. Firstly, Fig.4 is still not clear, please improve its clarity, and secondly,it seems better to move the content “4.1 Study limitations”  into  “5 conclusion”.

Author Response

Thank you very much for your comments and suggestions and the opportunity to improve our paper. Attached is a PDF that contains the detailed changes we did in out manuscript.

Reviewer 3 Report

thanks for the revision, but there are still errors in Figure 5-6, please check. Needs to be complemented by the managerial implications of the proposed recommendations

Author Response

We appreciate your comments and suggestions to improve our manuscript, thank you very much. Attached is a PDF file that contains detailed indication of the actions we took based on you feedback.
